# An Additive Chen Distribution with Applications to Lifetime Data

**Luis Carlos Méndez-González** \*,† , **Luis Alberto Rodríguez-Picón** † , **Ivan Juan Carlos Pérez-Olguín** and **Luis Ricardo Vidal Portilla**

Department of Industrial Engineering and Manufacturing, Autonomous University of Ciudad Juárez, Av. del Charro no. 450 Nte. Col. Partido Romero, Ciudad Juárez 32310, Mexico
* Correspondence: luis.mendez@uacj.mx
† These authors contributed equally to this work.

**Abstract:** This paper presents a lifetime model with properties representing increasing, decreasing, and bathtub curve shapes for failure rates. The proposed model was built based on the additive methodology, for which the Chen distribution was used as the base model, thus introducing the Additive Chen Distribution (AddC). An essential feature of AddC is this model's excellent flexibility in describing failure rates with non-monotonic behavior or with the shape of a bathtub curve concerning other current models. Statistical properties of AddC are presented and analyzed for different fields of study. For the estimation of AddC's parameters, the maximum likelihood method (MLE) was used. Three case studies in different fields of application are presented, from which AddC is compared against other probability distributions with similar properties. The results show that AddC offers competitive results.

**Keywords:** Additive Chen Distribution; bathtub distribution; Chen distribution; lifetime data analysis; non-monotone failure rate

**MSC:** 62N02; 62N05; 62P30





## 1. Introduction

Reliability analysis determines a device's behavior by the bathtub curve, which establishes the stages a product passes throughout its useful life. To verify whether the device follows a bathtub curve behavior, practitioners use statistical probability distributions and analyze the behaviors to determine essentials such as warranty and optimal maintenance times. Due to mathematical flexibility, classical probability distributions, such as the Weibull, Exponential or Lognormal distribution, are commonly used to describe this behavior. Nevertheless, this type of distribution cannot fit the data well if they have a non-monotonic behavior, such as the bathtub curve. The above causes uncertainty in the description of the information, and sometimes the conclusions of the analysis drop off due to the properties of certain distributions before different types of data.

Some researchers have proposed different methodologies to solve the problem of data representation with non-monotonic behavior. For example, refs. [1–7] developed models based on the additive methodology, which consists of adding the hazard functions of two statistical distributions, this to combine properties of the distributions. Other authors established the use of mathematical transformations to make the behavior of base distributions more flexible. By making the parameters of the distributions more flexible, it is possible to cover a more significant number of behaviors, including non-monotone. Examples of these works can be appreciated in [8–19]. Finally, another current of researchers has chosen to modify classical distributions such as the Weibull. The modifications presented consist of exponentiating parameters or adding a parameter to the distribution under analysis. These

modifications have been popular since they allow mathematical flexibility with which the analysis is simplified. Examples of these distributions can be seen in [20–25].

One of the main disadvantages of the work presented above is that the non-monotonic representation of the data resembles a "J" or "V" shape, which biases the complete information of the subject under analysis. This problem can be seen reflected in the design and quality of the product. Therefore, the device's behavior representation must be close to the bathtub curve and the designed specifications.

On the other hand, Chen [26] proposed distribution with two parameters, with which it is possible to represent data with a tendency to have incremental, decremental, or bathtub-shaped behavior. The hazard function of the Chen distribution (ChD) is defined as:

$$h_{ChD}(x) = \alpha \beta x^{(\beta-1)} e^{x^\beta},$$

(1)

where $\alpha$ and $\beta$ are the shape parameters. The ChD problem is that the representation of the failure rate is not symmetric to a bathtub shape, leading to the ChD being modified; this is because the ChD does not have a scale parameter that can make the distribution more flexible. For example, Thanh Thach and Briš [27], Tarvirdizade and Ahmadpour [28] presented the Additive Chen-Weibull (ACW), which combines the properties of the ChD and Weibull to obtain a hybrid model that offers very competitive results for survival analysis. Other studies proposed an extension of ChD based on transformations o generalizations; these works can be seen in [29–32]. The transformations mentioned above represent an alternative for reliability analysis but may not be a good option to determine the behavior of the lifetimes of the devices under analysis.

Therefore, this paper aims to present a new variant of the ChD, with which practitioners can represent the behavior of data with non-monotonic behavior in a way closer to that established by the bathtub curve. The proposed distribution is based on the additive methodology presented by Xie and Lai [1]. Thus, the Additive Chen Distribution (AddC) is preset and has two extra parameters to the base ChD. The extra parameters of AddC allow more flexibility for representing data with trends to be increasing, decreasing, or bathtub shaped. In turn, the MLE was used to estimate the parameters of the new distribution. On the other hand, to verify the properties of AddC, three case studies were designed, where AddC is compared against other distributions that have the property of modeling non-monotone or bathtub-shaped behavior. The Akaike information criterion (AIC), the Bayesian information criterion (BIC), the Kolmogorov Smirnov test (K-S), and the P-value were used to derive the conclusions of the case studies. The importance of AddC lies in offering competitive results concerning other distributions with similar properties, which helps practitioners to consider new analysis perspectives when performing reliability tests. For this new distribution, some functional statistical and mathematical properties are presented and discussed to demonstrate that AddC can be considered in real applications to determine the reliability of some devices' underperformance or acceleration life testing

Finally, this paper is organized as follows. Section 2 presents the general equations of AddC. Section 3 presents the Measures of Central Tendency of APD. Section 4 presents the moments and Incomplete Moments. Section 5 presents the order statistics. Section 6 presents the mean residual lifetime function. Section 7 presents the Rényi Entropy. Section 8 presents the Stress Strength Reliability. Section 9 presents the likelihood function to calculate the parameters proposed in Section 2. Section 10 presents the case studies of the paper. The last section provides concluding remarks and future work about the proposed model.

## 2. The New Lifetime Distribution

The additive methodology establishes that the hazard function of the new distribution, the sum of the hazard function of each distribution intended to be combined, must be made. Therefore, taking into account Equation (1) and retaking the same reparameterized equation to differentiate the ChDs, we obtain:

$$h(x) = h_{ChD1}(x) + h_{ChD2}(x),$$

where $h_{ChD1}(x) = \alpha\beta x^{(\beta-1)}e^{x^\beta}$ and $h_{ChD2}(x) = \lambda\theta x^{(\theta-1)}e^{x^\theta}$

Finally, the $h(x)$ of CPD is constituted by:

$$h(x) = \alpha\beta x^{(\beta-1)}e^{x^\beta} + \lambda\theta x^{(\theta-1)}e^{x^\theta}, \tag{2}$$

where, $\alpha, \lambda, \beta, \theta$ are the shape parameters of AddC with $\alpha, \beta, \lambda, \theta > 0$.

The Probability Density Function (PDF) of AddC from Equation (2) can be defined as:

$$f(x) = h(x) \cdot exp\left(-\int_0^x h(v)dv\right),$$
$$= \left(\alpha\beta x^{(\beta-1)}e^{x^\beta} + \lambda\theta x^{(\theta-1)}e^{x^\theta}\right) \cdot e^{\alpha\left(1-e^{x^\beta}\right)+\lambda\left(1-e^{x^\theta}\right)}. \tag{3}$$

The Survival Function $S(x)$ of AddC from Equations (2) and (3) can be written as:

$$S(x) = \frac{f(x)}{h(x)},$$
$$= e^{\alpha\left(1-e^{x^\beta}\right)+\lambda\left(1-e^{x^\theta}\right)}. \tag{4}$$

The Cumulative Density Function (CDF) $F(x)$ of AddC can be calculated as:

$$F(x) = 1 - S(x),$$
$$= 1 - \left(e^{\alpha\left(1-e^{x^\beta}\right)+\lambda\left(1-e^{x^\theta}\right)}\right). \tag{5}$$

Finally, the Cumulative Hazard Function of AddC $H(x)$ is defined as:

$$H(x) = -ln[S(x)],$$
$$= \alpha\left(e^{x^\beta}-1\right) + \lambda\left(e^{x^\theta}-1\right). \tag{6}$$

In Figures 1 and 2 you can see some forms that AddC takes for some random values.

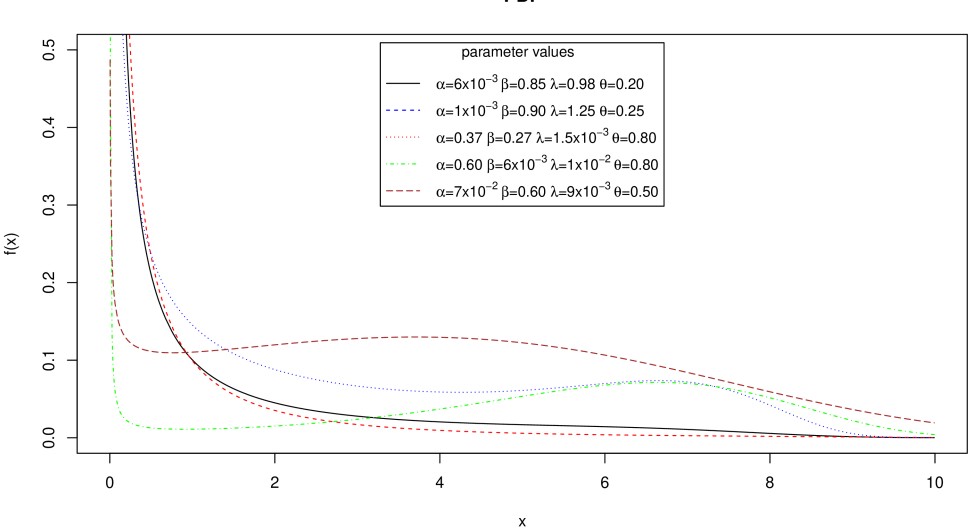

**Figure 1.** PDF for some parameter values derived of AddC.

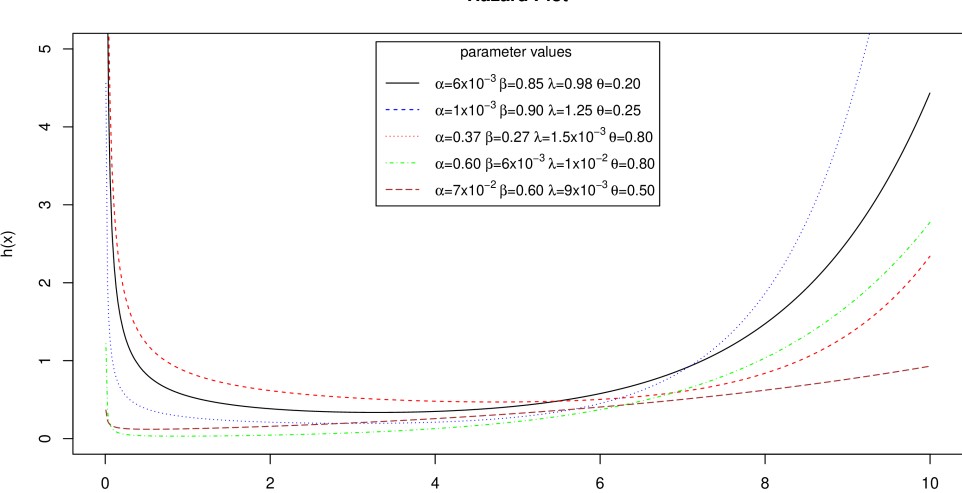

**Figure 2.** Hazard Rate for some parameter values derived of AddC.

## 3. Measures of Central Tendency

In this section, some statistical metrics of AddC are presented.

### 3.1. Quantile

The quantile is a useful equation in some branches of science to obtain random data with which it is possible to carry out PDF simulations. The *p*-th quantile $q_p$ denoted by *x* of AddC based on Equation (5), can be calculated as:

$$p = \left(1 - \left(e^{\alpha\left(1-e^{q^\beta}\right) + \lambda\left(1-e^{q^\theta}\right)}\right)\right). \tag{7}$$

For the case of AddC, Equation (7), does not have an analytical form. For this, numerical methods must be used to approximate solutions.

The median can be obtained from Equation (7) by setting $q = 0.5$.

### 3.2. Mode

The mode of AddC can be calculated by considering $f'(x) = 0$, so by taking Equation (3) and taking the first derivative; the following is obtained:

$$\frac{1}{x^2}e^{\alpha + \lambda\left(-e^{x^\theta}\right) + \lambda - \alpha e^{x^\beta}} \cdot \left[-\theta\lambda e^{x^\theta}x^\theta\left\{\theta\left(x^\theta\left\langle\lambda e^{x^\theta} - 1\right\rangle - 1\right) + 1\right\} + \alpha\beta e^{x^\beta}x^\beta\left(\beta - 2\theta\lambda e^{x^\theta}x^\theta - 1\right) - \alpha\beta^2 e^{x^\beta}x^{2\beta}\left(\alpha e^{x^\beta} - 1\right)\right] = 0. \tag{8}$$

As can be seen in Equation (8), the mode does not have an analytical solution, so as in the Quantile, it must be estimated by considering numerical methods.

## 4. Moments and Incomplete Moments

This section presents the formulation to obtain the equation of moments and incomplete moments, which are vital in lifetime models to determine the Mean Time To Failure (MTTF), the variance, and mean deviations between the mean and median.

### 4.1. Moments

The *rth* row moment of AddC is given by:

$$E(X^r) = \int_0^\infty rx^{r-1}S(x)dx.$$

$$= \int_0^\infty \left[ r x^{r-1} e^{\alpha\left(1-e^{x^\beta}\right)+\lambda\left(1-e^{x^\theta}\right)} \right] dx. \tag{9}$$

Applying series expansion, Equation (9) can be reduced as:

$$E(X^r) = \sum_{\imath,\jmath,k=0}^\infty \frac{r(-1)^\imath \alpha^\imath \imath^\jmath e^{(\alpha+\lambda)}}{\imath!\jmath!k!} \int_0^\infty \left( x^{(\jmath\beta+r-1)} e^{x^{\beta k}} \right) dx.$$

Finally, the moments of AddC can be calculated as:

$$E(X^r) = \sum_{\imath,\jmath,k=0}^\infty \frac{r(-1)^\imath \alpha^\imath \imath^\jmath e^{(\alpha+\lambda)}}{\imath!\jmath!k!} \Gamma\left( \frac{\beta\jmath+r}{\beta k} \right). \tag{10}$$

### 4.2. Incomplete Moments

The incomplete moments of AddC can be calculated from the following equation:

$$M_r(y) = \int_0^y x^r f(x) dx,$$

$$= \int_0^y x^r \left[ \left( \alpha\beta x^{(\beta-1)} e^{x^\beta} + \lambda\theta x^{(\theta-1)} e^{x^\theta} \right) \cdot e^{\alpha\left(1-e^{x^\beta}\right)+\lambda\left(1-e^{x^\theta}\right)} \right] dx. \tag{11}$$

Expanding by series Equation (11), the following is obtained:

$$M_r(y) = \sum_{\imath,\jmath,k,\ell=0}^\infty \frac{(-1)^{\imath+k} \alpha^\imath \imath^\jmath \lambda^k k^\ell e^{(\alpha+\lambda)}}{\imath!\jmath!k!\ell!} \int_0^y \left[ x^{r+\jmath\beta+\ell\theta} \left( \alpha\beta x^{\beta-1} e^{x^\beta} + \lambda\theta x^{\theta-1} e^{x^\theta} \right) \right] dx.$$

By solving the last integral, the incomplete moments of AddC can be expressed as:

$$M_r(y) = \sum_{\imath,\jmath,k,\ell=0}^\infty \frac{(-1)^{\imath+k} \alpha^\imath \imath^\jmath \lambda^k k^\ell e^{(\alpha+\lambda)}}{\imath!\jmath!k!\ell!} \cdot \Gamma\left( \frac{\beta\jmath + \lambda(\theta+1) + r}{\theta} \right). \tag{12}$$

## 5. Order Statistics

Order statistics are a popular representation in some branches of engineering, where redundancy blocks are used to prevent a system from stopping working. The order statistics are defined mathematically as:

$$f_{k:n}(x) = n \binom{n-1}{k-1} F(x)^{k-1} (1 - F(x))^{n-k} f(x). \tag{13}$$

By taking Equations (3) and (5) and replace them in Equation (13), the following is obtained:

$$f_{k:n}(x) = n \binom{n-1}{k-1} \left[ 1 - \left( e^{\alpha\left(1-e^{x^\beta}\right)+\lambda\left(1-e^{x^\theta}\right)} \right) \right]^{k-1} \tag{14}$$

$$\left[ e^{\alpha\left(1-e^{x^\beta}\right)+\lambda\left(1-e^{x^\theta}\right)} \right]^{n-k} \left[ \left( \alpha\beta x^{(\beta-1)} e^{x^\beta} + \lambda\theta x^{(\theta-1)} e^{x^\theta} \right) \cdot e^{\alpha\left(1-e^{x^\beta}\right)+\lambda\left(1-e^{x^\theta}\right)} \right]$$

By expanding Equation (14) by Taylor and Binomial series, the following is obtained:

$$= n \binom{n-1}{k-1} \left( \alpha\beta x^{(\beta-1)} e^{x^\beta} + \lambda\theta x^{(\theta-1)} e^{x^\theta} \right) \cdot \sum_{\imath=1}^{n-k} \left[ e^{\alpha\left(1-e^{x^\beta}\right)+\lambda\left(1-e^{x^\theta}\right)} \right]^\imath.$$

$$\left[1 - e^{\alpha\left(1-e^{x^\beta}\right)+\lambda\left(1-e^{x^\theta}\right)}\right]^{k-1} \cdot e^{\alpha\left(1-e^{x^\beta}\right)+\lambda\left(1-e^{x^\theta}\right)}$$

$$= n\binom{n-1}{k-1}\left(\alpha\beta x^{(\beta-1)}e^{x^\beta} + \lambda\theta x^{(\theta-1)}e^{x^\theta}\right) \cdot \sum_{\iota=1}^{n-k}\left[e^{\alpha\left(1-e^{x^\beta}\right)+\lambda\left(1-e^{x^\theta}\right)}\right]^\iota \cdot$$

$$\sum_{j=1}^{k-1}(-1)^j(k-1)^j\left[e^{\alpha\left(1-e^{x^\beta}\right)+\lambda\left(1-e^{x^\theta}\right)}\right]^j$$

Finally, the k-out-n state of AddC can be described as:

$$f_{k:n}(x) = n\binom{n-1}{k-1}h(x) \cdot \sum_{\iota=1}^{n-k}\sum_{j=1}^{k-1}\left[(-1)^j S(x)^{\iota+j}(k-1)^j\right] \tag{15}$$

## 6. Mean Residual Lifetime

In survival or reliability analysis, the mean residual lifetime (MRL) is essential to characterize the models. Thus, the expected additional lifetime in life testing scenarios has given that a component system that has survived until time $x$ is called the MRL [33]. The MRL is defined as:

$$M(t) = E(X - t|X > t) = \frac{1}{S(t)}\int_t^\infty S(x)dx = \frac{1}{S(t)}\int_0^\infty S(x+t)dx. \tag{16}$$

By taking Equation (4) and replace it into Equation (16), the following is obtained:

$$M(t) = \frac{1}{e^{\alpha\left(1-e^{t^\beta}\right)+\lambda\left(1-e^{t^\theta}\right)}}\int_0^\infty\left[e^{\alpha\left(1-e^{(x+t)^\beta}\right)+\lambda\left(1-e^{(x+t)^\theta}\right)}\right]dx.$$

Developing by series and solving the integral, the MRL of AddC is expressed as:

$$M(t) = \frac{e^{-\alpha\left(1+e^{t^\beta}\right)-\lambda\left(1+e^{t^\theta}\right)}}{\theta}\sum_{\iota,j=0}^\infty\frac{\lambda^\iota\iota^j}{\iota!j!\lambda^{\iota\beta+1}}\Gamma\left(\frac{j\beta+1}{\theta}\right). \tag{17}$$

## 7. Entropy

One way to quantify the uncertainty or randomness of systems is through entropy. For AddC, we will formulate the Rényi entropy, which is defined as:

$$I_R(\delta) = \frac{1}{1-\delta} \cdot log\int_0^\infty f(x)^\delta dx. \tag{18}$$

By taking Equation (3) and substituting in Equation (18), the Rényi Entropy for AddC can be calculated as follows:

$$I_R(\delta) = \frac{1}{1-\delta} \cdot log\int_0^\infty\left[\left(\alpha\beta x^{(\beta-1)}e^{x^\beta} + \lambda\theta x^{(\theta-1)}e^{x^\theta}\right) \cdot e^{\alpha\left(1-e^{x^\beta}\right)+\lambda\left(1-e^{x^\theta}\right)}\right]^\delta dx.$$

Applying the Taylor series expansion to the function $f(x)^\delta$, the following is obtained:

$$I_R(\delta) = \frac{1}{1-\delta} \cdot log\sum_{\iota=0}^\delta\sum_{j,k,\ell,\mu,\nu,\rho,\tau}^0\binom{\delta}{\iota}\frac{(\delta-1)^\iota(\alpha\beta)^j(\lambda\theta)^\ell(\delta\alpha)^\mu\mu^\nu(-\lambda\delta)^\rho\rho^\tau}{\mu!\nu!\rho!\tau!}$$

$$\cdot\int_0^\infty\left(x^{j+\beta(\mu+1)+\theta(k+\rho+1)+\ell-2}e^{\iota x^\theta}\right)dx.$$

By solving the last integral, the Rényi Entropy of AddC can be written as:

$$I_R(\delta) = \frac{1}{1-\delta} \cdot \log \sum_{\iota=0}^{\delta} \sum_{j,k,\ell,\mu,\nu,\rho,\tau}^{0} \binom{\delta}{\iota} \frac{(\delta-1)^{\iota}(\alpha\beta)^{j}(\lambda\theta)^{\ell}(\delta\alpha)^{\mu}\mu^{\nu}(-\lambda\delta)^{\rho}\rho^{\tau}}{\mu!\nu!\rho!\tau!} \cdot$$
$$\Gamma\left(\frac{\beta(\mu+1)+\theta(\mu+k+1)+\ell}{\theta}\right) \tag{19}$$

## 8. Stress-Strength Reliability

This equation is primarily used in reliability engineering to determine the probability of a component performing a specified function under a stated stress condition in a specified scenario. The Stress-Strength (SS) is defined as follows:

$$R = P(X_1 > X_2) = \int_0^\infty fx_1(x) \cdot Fx_2(x)dx. \tag{20}$$

Taken Equations (3) and (5) and following Equation (20), the SS can be obtained as:

$$R = \int_0^\infty \left( \alpha_1\beta_1 x_1^{(\beta_1-1)} e^{x_1^{\beta_1}} + \lambda_1\theta_1 x_1^{(\theta_1-1)} e^{x_1^{\theta_1}} \right) \cdot e^{\alpha_1\left(1-e^{x_1^{\beta_1}}\right)+\lambda\left(1-e^{x_1^{\theta_1}}\right)}$$
$$\cdot \left\{ 1 - \left( e^{\alpha_2\left(1-e^{x_2^{\beta_2}}\right)+\lambda_2\left(1-e^{x_2^{\theta_2}}\right)} \right) \right\} dx.$$

Expanding by series and reducing terms by mathematical means the SS for AddC can be written as:

$$R = \sum_{\iota,j,k,\ell=0}^{\infty} \sum_{m,\eta,o,\rho=0}^{\infty} \frac{(-1)^{\iota+k+m+o}\alpha_1^{\iota}\alpha_2^{m}\iota^{j}m^{\eta}\lambda_1^{k}k^{\ell}o^{\rho}\lambda_2^{o} e^{\alpha_1+\theta_1-(\alpha_2+\theta_2)}}{\iota!j!k!\ell!m!\eta!o!\rho!} \cdot$$
$$\Gamma\left(\frac{\eta\beta_2+\beta_1(j+1)+\ell\theta_1+\rho\theta_2}{\theta_1\beta_2}\right). \tag{21}$$

## 9. Maximum Likelihood Estimator

A primordial aspect of statistical models is the formulation of a method that will estimate the values of each parameter in the distribution. For the case of AddC, the use of the MLE is proposed. Let $X_1, X_2 \ldots X_m$ be a random sample from the lifetime distribution with PDF $f(x)$ and based on a sample of size $m$. Then the likelihood function of Equation (3) is written as follows:

$$L = \prod_{\iota=1}^{m} \left[ \left( \alpha\beta x^{(\beta-1)} e^{x^\beta} + \lambda\theta x^{(\theta-1)} e^{x^\theta} \right) \cdot e^{\alpha\left(1-e^{x^\beta}\right)+\lambda\left(1-e^{x^\theta}\right)} \right].$$

If we compute the natural logarithm of the above equation, the log-likelihood ($\Lambda$) function becomes:

$$\Lambda = m(\alpha+\lambda) + \sum_{\iota=1}^{m}\left[\ln\left(\alpha\beta x_i^{\beta-1}e^{x_i^\beta}+\lambda\theta x_i^{\theta-1}e^{x_i^\theta}\right)\right] + \sum_{\iota=1}^{m}\left[-\alpha\, e^{x_i^\beta}-\lambda\, e^{x_i^\theta}\right]. \tag{22}$$

By taking the first partial derivative of Equation (22), the estimation for each parameter $\alpha, \beta, \lambda$, and $\theta$ can be calculated as follow:

$$\frac{\partial\Lambda}{\partial\alpha} = m + \sum_{\iota=1}^{m}\left(\frac{\beta x_i^\beta e^{x_i^\beta}}{\alpha\beta x_i^\beta e^{x_i^\beta}+\lambda\theta x_i^\theta e^{x_i^\theta}} - e^{x_i^\beta}\right). \tag{23}$$

$$\frac{\partial \Lambda}{\partial \beta} = \sum_{\iota=1}^{m} \left( \frac{\alpha\beta x_i^{\beta-1}\ln(x_i)\mathrm{e}^{x_i^{\beta}} + \alpha\beta x_i^{2\beta-1}\ln(x_i)\mathrm{e}^{x_i^{\beta}} + \alpha x_i^{\beta-1}\mathrm{e}^{x_i^{\beta}}}{\alpha\beta x_i^{\beta-1}\mathrm{e}^{x_i^{\beta}} + \lambda\theta x_i^{\theta-1}\mathrm{e}^{x_i^{\theta}}} - \alpha x_i^{\beta}\ln(x_i)\mathrm{e}^{x_i^{\beta}} \right). \tag{24}$$

$$\frac{\partial \Lambda}{\partial \theta} = \sum_{\iota=1}^{m} \left( \frac{\lambda\theta x_i^{\theta-1}\ln(x_i)\mathrm{e}^{x_i^{\theta}} + \lambda\theta x_i^{2\theta-1}\ln(x_i)\mathrm{e}^{x_i^{\theta}} + \lambda x_i^{\theta-1}\mathrm{e}^{x_i^{\theta}}}{\alpha\beta x_i^{\beta-1}\mathrm{e}^{x_i^{\beta}} + \lambda\theta x_i^{\theta-1}\mathrm{e}^{x_i^{\theta}}} - \lambda x_i^{\theta}\ln(x_i)\mathrm{e}^{x_i^{\theta}} \right). \tag{25}$$

$$\frac{\partial \Lambda}{\partial \lambda} = m + \sum_{\iota=1}^{m} \left( \frac{\theta x_i^{\theta-1}\mathrm{e}^{x_i^{\theta}}}{\alpha\beta x_i^{\beta-1}\mathrm{e}^{x_i^{\beta}} + \lambda\theta x_i^{\theta-1}\mathrm{e}^{x_i^{\theta}}} - \mathrm{e}^{x_i^{\theta}} \right). \tag{26}$$

From Equations (23)–(26), we can derive the Fisher observation matrix, with which it is possible to obtain the Hessian. In Appendix A, the elements of the Fisher matrix are presented.

## 10. Case of Study

In this section, the behavior of AddC is analyzed in three case studies where the data has non-monotonic behavior. To verify the methodology established in this paper, a comparative study was designed between AddC and other distributions with similar properties. The distributions considered were the ACW proposed by Thanh Thach and Briš [27], Tarvirdizade and Ahmadpour [28], the Additive Weibull Distribution (AWD) proposed by Xie and Lai [1], the Additive Perks-Weibull (APW) proposed by Singh [5], Perks4 proposed by Zeng et al. [34], Modified Weibull Distribution (MW) proposed by Al-malki and Yuan [22], and the Sarhan and Zaindin modified Weibull (SZMW) proposed by Zaindin and Sarhan [23]. Table 1 shows the hazard functions of each distribution used for the comparative analysis.

**Table 1.** Bathtub Shape Distributions.

| Model | $h(x)$ |
|---|---|
| ACW | $\alpha\beta x^{\beta-1}e^{x^{\beta}} + \lambda\theta x^{\theta-1}$ |
| APW | $\frac{\alpha\lambda e^{\lambda x}}{1+\alpha e^{\lambda x}} + \theta\beta x^{\beta} - 1$ |
| ADW | $\alpha\lambda x^{\lambda-1} + \beta\theta x^{\theta-1}$ |
| Perks4 | $\frac{\theta + e^{(\beta x+\alpha)}}{1 - e^{(-\beta x+\lambda)}}$ |
| MW | $\beta(\alpha + \lambda t)x^{\alpha-1}e^{\lambda x}$ |
| SZMW | $\alpha + \beta\lambda x^{\lambda-1}$ |

In all of the proposed case studies, the parameters were estimated through the MLE. This algorithm was programmed through R using the MaxLik package [35]. In turn, the AIC, BIC, K-S, and P-Value were calculated for each distribution to establish the discussion and draw conclusions from the results.

### 10.1. Case Study 1: Lifetime of 50 Devices

This case study focuses on determining the behavior of the failure times of 50 devices. The data was obtained by Aarset [36] and is presented in Table 2.

**Table 2.** Aarset Data of 50 Devices.

| DATA | | | | | | | | | |
|---|---|---|---|---|---|---|---|---|---|
| 0.1 | 0.2 | 1 | 1 | 1 | 1 | 1 | 2 | 3 | 6 |
| 7 | 11 | 12 | 18 | 18 | 18 | 18 | 18 | 21 | 32 |
| 36 | 40 | 45 | 46 | 47 | 50 | 55 | 60 | 63 | 63 |
| 67 | 67 | 67 | 67 | 72 | 75 | 79 | 82 | 82 | 83 |
| 84 | 84 | 84 | 85 | 85 | 85 | 85 | 85 | 86 | 86 |

To identify the behavior of the data, the total test time (TTT) curve is used. Therefore, in Figure 3, the TTT of the data presented in Table 2 is presented.

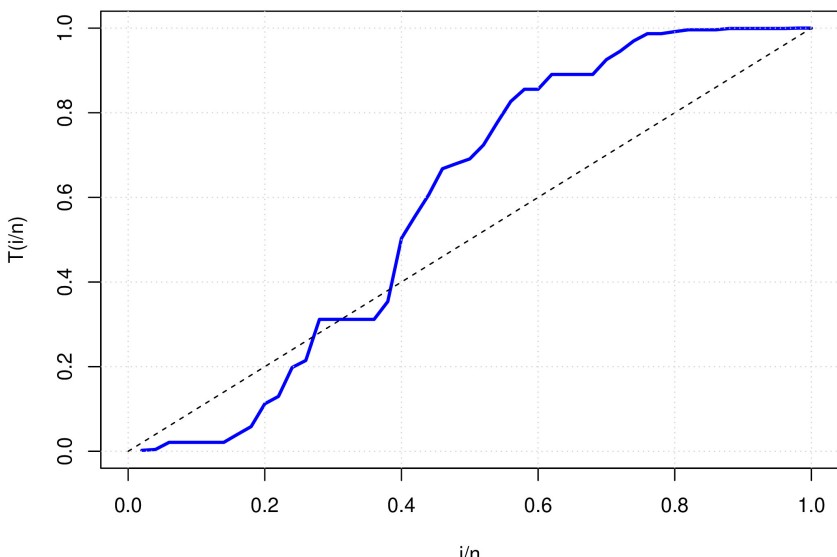

**Figure 3.** TTT plot for Data presented in Table 2.

The results shown in Figure 3, show that the data presented in Table 2 exhibit a non-monotonic behavior.

On the other hand, Table 3 shows the estimation of each parameter for each of the models presented in Table 1 and AddC. The information presented in Table 3 shows statistical evidence that AddC offers a better description of the behavior of device failure times concerning the distributions listed in Table 1. The previous is based on the AIC and BIC criteria, where AddC has the smallest value of these criteria concerning the other distributions compared within the analysis carried out, which indicates that AddC may be a good candidate for the description of the data in Table 2. On the other hand, AddC shows a better fit in the data representation, based on the P-value described in Table 3. This P-value represents that the data will be as close as possible to the device's behavior under the operating conditions that the manufacturer has described.

In turn, with the information obtained in Table 3, it is possible to make a graphic description of the behavior of the devices over time, for which Figure 4 shows the representation of the PDF, Reliability, Hazard, and Cumulative Hazard.

**Table 3.** Estimated Values, standard errors in brackets and Statistics metrics for the Case of Study 1.

| Model | Parameters | | | | Statistics | | | | |
|---|---|---|---|---|---|---|---|---|---|
| | $\alpha$ | $\beta$ | $\lambda$ | $\theta$ | Loglik | AIC | BIC | K-S | *p*-Value |
| AddC | $5.977 \times 10^{-2}(5.958 \times 10^{-4})$ | $0.249(7.114 \times 10^{-3})$ | $2.082 \times 10^{-17}(1.146 \times 10^{-22})$ | $0.822(1.654 \times 10^{-2})$ | $-203.054$ | 414.108 | 421.756 | 0.060 | 0.975 |
| ACW | $4.215 \times 10^{-2}(1.021 \times 10^{-5})$ | $0.278(2.224 \times 10^{-2})$ | $1.1183 \times 10^{-2}(3.995 \times 10^{-4})$ | $86.231(2.414)$ | $-205.350$ | 418.710 | 426.360 | 0.070 | 0.965 |
| APW | $7.158 \times 10^{-17}(1.112 \times 10^{-19})$ | $0.688(1.025 \times 10^{-2})$ | $0.443(1.921 \times 10^{-2})$ | $5.320 \times 10^{-2}(4.221 \times 10^{-2})$ | $-212.870$ | 433.750 | 441.440 | 0.091 | 0.804 |
| ADW | $8.448 \times 10^{-9}(1.224 \times 10^{-10})$ | $0.091(3.814 \times 10^{-2})$ | $4.279(4.911 \times 10^{-2})$ | $0.466(9.814 \times 10^{-2})$ | $-221.350$ | 450.712 | 458.360 | 0.127 | 0.393 |
| Perks4 | $-71.432(9.124)$ | $0.839(0.111)$ | $-6.211 \times 10^{-2}(1.321 \times 10^{-4})$ | $1.534 \times 10^{-2}(1.194 \times 10^{-3})$ | $-205.610$ | 419.112 | 424.824 | 0.079 | 0.901 |
| MW | $0.355(0.115)$ | $6.221 \times 10^{-2}(2.701 \times 10^{-2})$ | $2.311 \times 10^{-2}(5 \times 10^{-3})$ | $-$ | $-227.150$ | 460.310 | 464.045 | 0.134 | 0.334 |
| SZMW | $1.311 \times 10^{-2}(3.014 \times 10^{-3})$ | $3.808 \times 10^{-9}(6.805 \times 10^{-10})$ | $4.405(0.145)$ | $-$ | $-229.410$ | 464.821 | 470.560 | 0.151 | 0.202 |

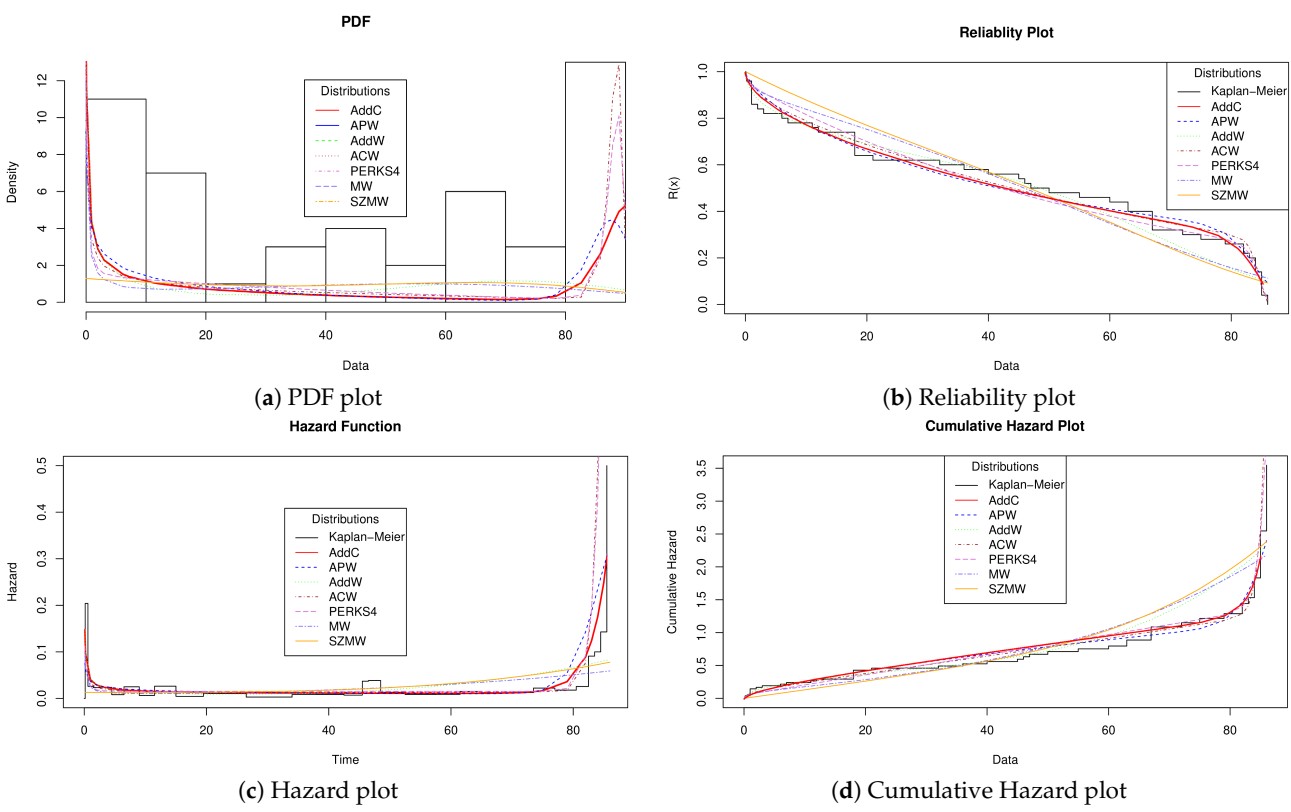

**(a)** PDF plot

**(b)** Reliability plot

**(c)** Hazard plot

**(d)** Cumulative Hazard plot

**Figure 4.** Reliability Plots for Lifetime data presented in Table 2.

Figure 4a shows the PDF behavior for each distribution under analysis by considering how the data histogram in Table 2, Perks4, AWC, APW, and AddC offer a good representation of the distribution of lifetimes. Although the above does not represent convincing evidence, it can give an idea of which distributions can offer behaviors close to the reality of the device under analysis. Figure 4b shows the behavior of the lifetimes of Table 2 concerning the reliability equation of each distribution under analysis. The graph mentioned above is used within reliability engineering to determine essential aspects such as determining the estimated lifetime of the device under analysis or maintenance times to extend the product's life. For the specific case and considering the Kaplan–Meier empirical reliability, AddC offers a better fit concerning the other distributions used in the comparative analysis. That means AddC will more accurately describe lifetimes, maintenance times, and MTTF; those metrics are essential in reliability analysis. Figure 4c shows the form taken by the failures of the device under analysis. As seen in the vast majority of the distributions, a behavior similar to the bathtub curve can be observed, which confirms the empirical behavior obtained by the TTT plot (see Figure 3). Now, suppose we visually contrast the behavior of the distributions and compare it against the results of Table 3. In that case, we can conclude that AddC touches the empirical behavior in more points, which indicates that AddC offers a more competitive representation of failures within the device. Product engineers can consider the above information to improve designs, reduce manufacturing, and estimate maintenance costs. Finally, Figure 4d shows the behavior of the accumulated hazard. In this graph, it can be seen that AddC is closer to the Kaplan–Meier empirical representation. The preceding indicates that AddC will better model the failures of the device as its useful life elapses.

### 10.2. Case of Study 2: Field-Tracking Study of a Larger System

This case study analyzed the behavior of 30 devices inserted into the primary electrical system after a series of determined conditions and time had elapsed. These data were presented by Meeker et al. [37]. In turn, the failures were classified into two types; failures

due to wear or electrical variation produced by the electrical source. The result of the failure cycles are presented in Table 4.

**Table 4.** Meeker and Escobar Data of 30 Devices.

| DATA | | | | | | | | | |
|------|------|------|------|------|------|------|------|------|------|
| 275 | 13 | 147 | 23 | 181 | 30 | 65 | 10 | 300 | 173 |
| 106 | 300 | 300 | 212 | 300 | 300 | 300 | 2 | 261 | 293 |
| 88 | 247 | 28 | 143 | 300 | 23 | 300 | 80 | 245 | 266 |

The TTT plot of the data presented in Table 4 is presented in Figure 5.

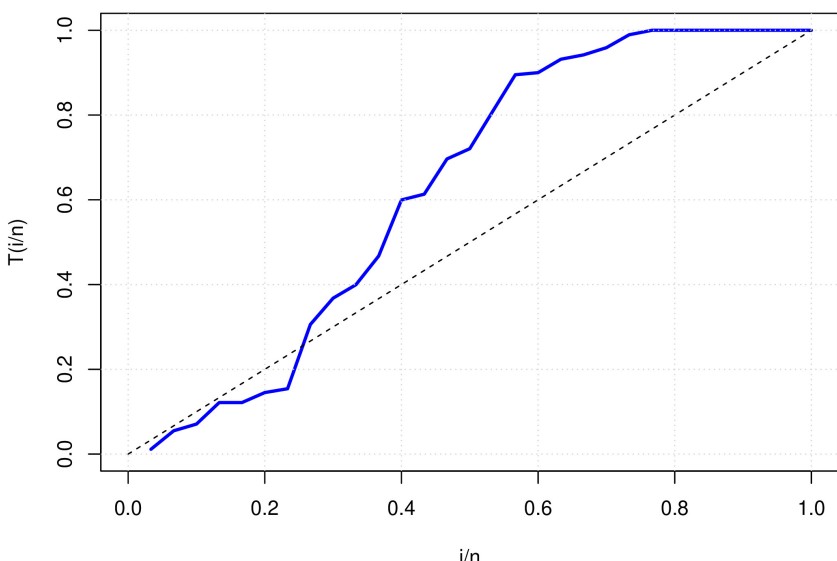

**Figure 5.** TTT plot for Data presented in Table 4.

The form that the data takes, as established in the TTT plot of Figure 5, indicates that the data of Table 4 have a behavior similar to that of a bathtub curve. Therefore, it is submitted to the analysis with the distributions established in Table 1. Table 5 shows the results obtained from the estimates of each parameter for the models under analysis.

Table 5 shows that AddC offers a better fit than the other distributions under analysis. However, the adjustment of AddC is not the best since the P-Value reaches a value above 0.7, which is possible that for this specific case, AddC may not represent all the data presented in Table 4. Those mentioned above can be better exposed when representing the behavior of the estimates and the data of this case study, so Figure 6 shows the behavior graphs of the data presented in Table 4.

Figure 6a shows the behavior of the PDF for each of the distributions established in the comparative analysis. As can be seen, AddC represents more competitively the behavior of the data density. The previous offer AddC a competitive advantage that reliability practitioners make decisions about what kind of distributions they can use to obtain analyzes closer to the reality of the product they are analyzing. In Figure 6b, the behavior of the distributions' reliability curve is observed and compared concerning the Kaplan–Meier representation. The conclusion obtained from Figure 6b is that AddC passes the Kaplan–Meier representation through a more significant number of points concerning the distributions listed in Table 1; This means that AddC can get closer to the device operation times such as warranty times, MTTF and maintenance times. These times are vital in the design of the product since, with these times, it is possible to establish substantial improvements.

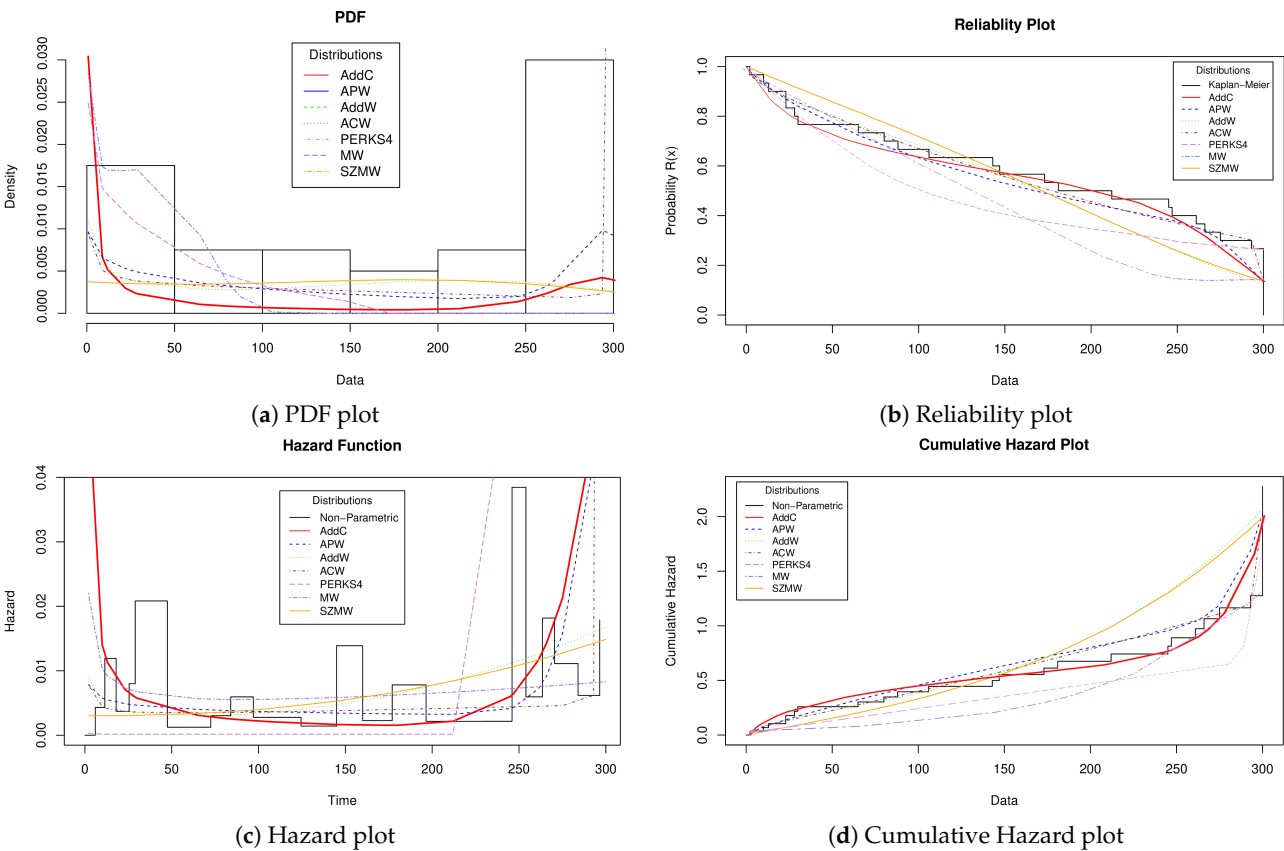

**Figure 6.** Reliability Plots for Lifetime data presented in Table 4.

On the other hand, Figure 6c shows the behavior of the hazard function of failure times, where it is notable that despite the data described in Table 4 being known to behave similarly to a bathtub curve. Some distributions that possess the above property cannot reliably represent non-monotonic behavior. The above described leads to an essential loss of information in describing the behavior of the device under analysis, so practitioners must consider this when choosing the distribution that will describe the failure times.

The AddC shows a very competitive fit concerning the non-parametric representation of the data presented in Table 4 and the distributions presented in Table 1. So It can be concluded that AddC is the best option to describe the failure times in this case study. Finally, Figure 6d shows the behavior of the cumulative hazard. As in the previous cases, if we compare each distribution's form concerning the non-parametric representation. We can observe that AddC better adjusts the faults accumulated in the device, which gives an idea more precise to the practitioners of how it is that the faults are presented throughout the useful life of the device under analysis.

**Table 5.** Estimated Values, standard errors in brackets and Statistics metrics for the Case of Study 2.

| Model | Parameters | | | | Statistics | | | | |
|---|---|---|---|---|---|---|---|---|---|
| | $\alpha$ | $\beta$ | $\lambda$ | $\theta$ | Loglik | AIC | BIC | K-S | *p*-Value |
| AddC | $3.042 \times 10^{-11}(5.958 \times 10^{-13})$ | $0.561(7.114 \times 10^{-3})$ | $0.407(1.146 \times 10^{-2})$ | $8.419 \times 10^{-2}(1.654 \times 10^{-3})$ | $-147.887$ | 303.774 | 311.422 | 0.112 | 0.714 |
| ACW | $1.518 \times 10^{-2}(2.101 \times 10^{-3})$ | $0.260(3.014 \times 10^{-3})$ | $3.331 \times 10^{-3}(1.741 \times 10^{-5})$ | $259.427(14.558)$ | $-151.340$ | 310.670 | 316.280 | 0.132 | 0.652 |
| APW | $5.142 \times 10^{-12}(8.063 \times 10^{-8})$ | $0.807(0.172)$ | $8.802 \times 10^{-2}(2.114 \times 10^{-3})$ | $1.114 \times 10^{-2}(9.332 \times 10^{-3})$ | $-167.910$ | 343.820 | 349.420 | 0.134 | 0.655 |
| ADW | $6.915 \times 10^{-9}(9.867 \times 10^{-10})$ | $1.803 \times 10^{-2}(1.788 \times 10^{-2})$ | $3.349(5.422 \times 10^{-2})$ | $0.642(0.196)$ | $-176.97$ | 361.940 | 367.540 | 0.163 | 0.401 |
| Perks4 | $-95.721(11.665)$ | $0.579(0.111)$ | $-2.421 \times 10^{-3}(1.025 \times 10^{-4})$ | $1.534 \times 10^{-2}(1.194 \times 10^{-3})$ | $-180.225$ | 368.450 | 374.054 | 0.185 | 0.251 |
| MW | $0.495(0.228)$ | $6.221 \times 10^{-2}(2.701 \times 10^{-2})$ | $2.311 \times 10^{-2}(5 \times 10^{-3})$ | $-$ | $-178.160$ | 362.330 | 366.530 | 0.168 | 0.362 |
| SZMW | $3.012 \times 10^{-3}(1.025 \times 10^{-3})$ | $8.853 \times 10^{-9}(2.955 \times 10^{-9})$ | $3.266(6.241 \times 10^{-2})$ | $-$ | $-178.470$ | 362.940 | 367.140 | 0.175 | 0.316 |

### 10.3. Case of Study 3: Lifetime of 18 Electronic Devices

The following case study is based on the results obtained by Wang [7], who tested 18 electronic devices. The failure times obtained are shown in Table 6.

**Table 6.** Wang [7] Data of 18 electronic devices.

| | | | | Data | | | | |
|---|---|---|---|---|---|---|---|---|
| 5 | 11 | 21 | 31 | 46 | 75 | 98 | 122 | 145 |
| 165 | 196 | 224 | 245 | 293 | 321 | 330 | 350 | 420 |

The TTT plot is obtained to verify the behavior of the data presented in Table 6. Therefore, Figure 7 shows the behavior of the data presented by Wang [7].

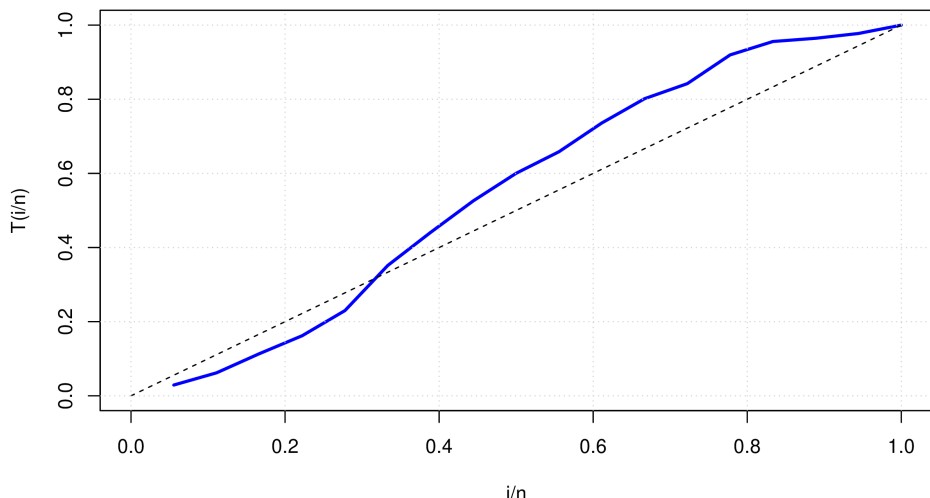

**Figure 7.** TTT plot for Data presented in Table 6.

As shown in Figure 7, empirically, the TTT plot indicates that the data presented in Table 6 show behavior resembling the bathtub curve. Therefore, the estimation of the parameters is carried out through the MLE for all the distributions placed under analysis. Table 7 presents the results of the parameters' estimations and the statistical decision criteria.

Based on the evidence shown in Table 7, it can be established that AddC has good results for describing the data established in Table 6. Although, it should be noted that in this case study, some of the distributions established for the analysis offer very competitive results, where the difference is minimal concerning AddC. Nevertheless, this slight difference can establish a decision criterion regarding which distribution to use in the analysis.

The graphical representation of the results obtained in Table 7 is shown in Figure 8.

As in the two previous case studies, Figure 8 shows the device's behavior under the different reliability representations. The representation of the PDF is shown in Figure 8a, where it can be seen that the distributions under analysis offer a good representation of the histogram of the data presented in Table 6. So, in this case, the PDF cannot draw a satisfactory conclusion about which distribution can be used practically. The reliability graph presented in Figure 8b shows that the adjustment of AddC touches the Kaplan–Meier representation in a more significant number of points. This aspect shows that AddC could express closer to the operating environments the behavior of the device under analysis.

**Table 7.** Estimated Values, standard errors in brackets and Statistics metrics for the Case of Study 3.

| Model | Parameters | | | | Statistics | | | | |
|---|---|---|---|---|---|---|---|---|---|
| | $\alpha$ | $\beta$ | $\lambda$ | $\theta$ | Loglik | AIC | BIC | K-S | *p*-Value |
| AddC | $1.946 \times 10^{-7}(1.642 \times 10^{-6})$ | $0.460(8.594 \times 10^{-2})$ | $1.576 \times 10^{-2}(1.266 \times 10^{-2})$ | $0.263(3.619 \times 10^{-2})$ | $-107.584$ | 223.168 | 226.729 | 0.045 | 0.997 |
| ACW | $6.884 \times 10^{-2}(3.225 \times 10^{-3})$ | $0.228(5.332 \times 10^{-2})$ | $4.638 \times 10^{-2}(6.452 \times 10^{-3})$ | $0.224(1.445 \times 10^{-2})$ | $-116.738$ | 241.477 | 245.039 | 0.378 | 0.325 |
| APW | $1.012 \times 10^{-4}(2.014 \times 10^{-3})$ | $0.799(0.325)$ | $2.722 \times 10^{-2}(4.321 \times 10^{-2})$ | $1.322 \times 10^{-2}(1.722)$ | $-108.201$ | 224.402 | 227.963 | 0.059 | 0.994 |
| ADW | $4.361 \times 10^{-7}(4.058 \times 10^{-6})$ | $1.522 \times 10^{-2}(2.145 \times 10^{-2})$ | $4.025 \times 10^{-3}(2.025 \times 10^{-3})$ | $2.557(0.545)$ | $-108.880$ | 225.760 | 229.320 | 0.092 | 0.991 |
| Perks4 | $-9.282(2.225)$ | $1.441 \times 10^{-2}(2.225 \times 10^{-2})$ | $-0.422(1.022 \times 10^{-2})$ | $2.821 \times 10^{-3}(3.221 \times 10^{-5})$ | $-108.125$ | 224.250 | 227.811 | 0.095 | 0.995 |
| MW | $0.646(0.309)$ | $1.522 \times 10^{-2}(2.012 \times 10^{-2})$ | $4.012 \times 10^{-3}(2.114 \times 10^{-3})$ | $-$ | $-108.935$ | 223.860 | 226.540 | 0.094 | 0.992 |
| SZMW | $2.311 \times 10^{-3}(7.012 \times 10^{-3})$ | $6.211 \times 10^{-4}(5.114 \times 10^{-3})$ | $1.306(1.113)$ | $-$ | $-110.340$ | 226.680 | 229.350 | 0.107 | 0.971 |

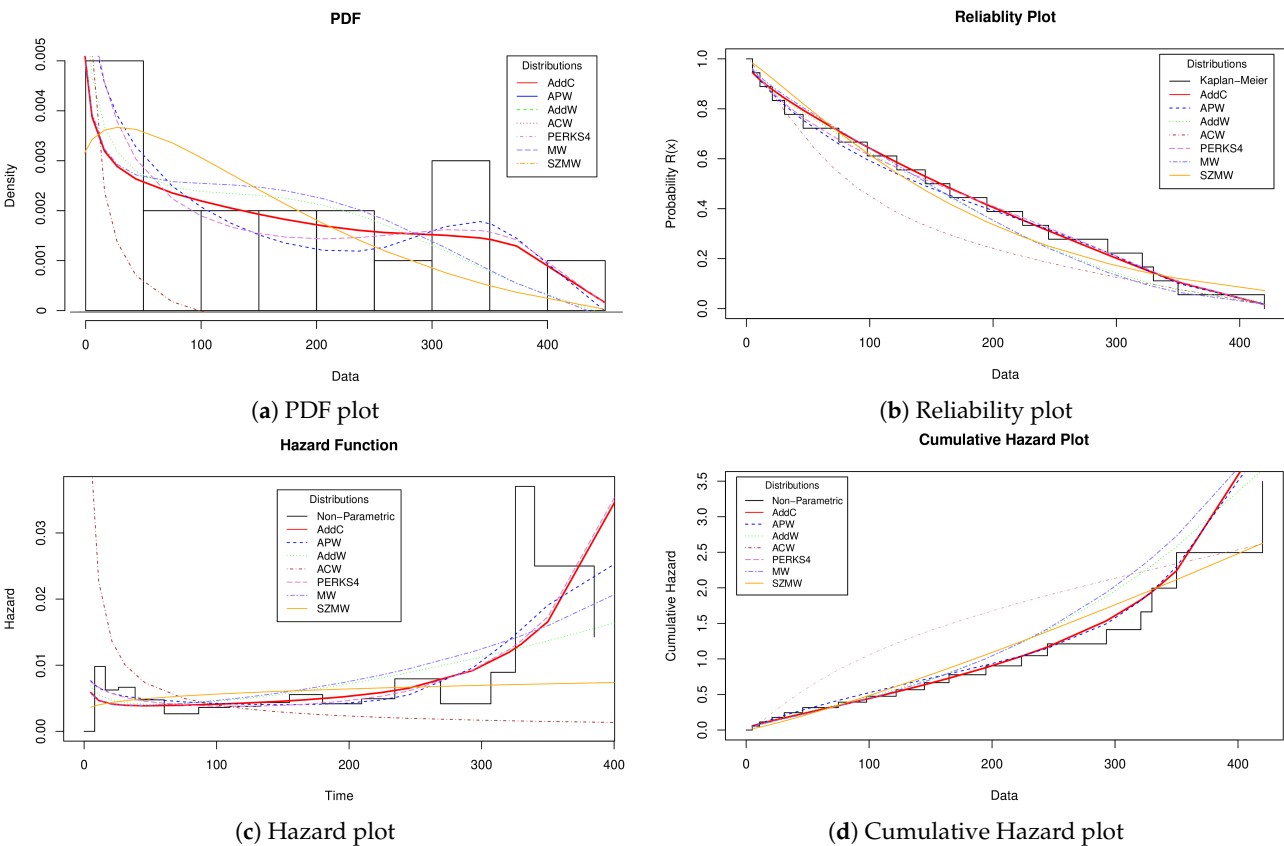

**Figure 8.** Reliability Plots for Lifetime data presented in Table 6.

The shape taken by the failure times in Figure 8c shows that AddC manages to get closer to the shape of the bathtub curve. In turn, the other distributions listed in Table 1 exhibit a non-monotonous behavior but without a defined bathtub curve shape. In this case, AddC could be a good choice for practitioners of reliability engineering.

Finally, Figure 8d shows the shape of the cumulative hazard plot of each distribution included in the analysis. In this case, we can see that AddC and the APW have a good fit, but AddC shows slightly closer to the non-parametric representation, with which AddC can establish the representation of device faults.

## 11. Conclusions

In this paper, a statistical distribution with applications to reliability engineering for the description of device failure times was presented. The distribution was based on the Chen distribution and the additive methodology, thus introducing AddC, with two shape and scale parameters. The AddC has the property of competitively describing non-monotonic behaviors, especially in the form of a bathtub curve, which theoretically represents the useful life of any device.

The statistical properties analyzed show the flexibility that AddC can have in different analysis scenarios, which allows AddC to approach describing the real behavior of the devices under analysis. In turn, it allows reliability practitioners to determine essential aspects in the analysis, such as warranty and maintenance times. With this information, it is possible to improve device designs and reduce manufacturing costs. The assumptions or conditions for the use of AddC are similar to those of any probability distribution. So the results that AddC yields are based on the quality of the data put under analysis; therefore, there are no special conditions for AddC application.

The competitiveness of AddC was tested on three case studies designed where the failure times of the devices are non-monotonic. In all case studies, the results obtained from AddC were compared against other statistical distributions with similar mathematical

properties. This comparative study aimed to demonstrate that AddC can be a good option for reliability analysis in different devices.

In future work to extend the applications of AddC, Bayesian techniques are proposed to know the differences between the results obtained by MLE and the Bayes techniques. On the other hand, AddC can be modified to be used in accelerated life tests, which is one of the most used techniques in reliability for obtaining information in short times through the acceleration of some variable that directly affects the life of the devices under analysis.

**Author Contributions:** Conceptualization, L.C.M.-G.; methodology, L.C.M.-G.; validation, L.A.R.-P. and I.J.C.P.-O.; data curation, L.A.R.-P. and I.J.C.P.-O.; formal analysis, L.C.M.-G. and L.A.R.-P.; investigation, L.C.M.-G. and L.R.V.P.; supervision, L.C.M.-G. and L.R.V.P.; resources, L.C.M.G; writing—original draft preparation, L.C.M.-G.; writing—review and editing, L.A.R.-P., I.J.C.P.-O. and L.R.V.P.; visualization, L.A.R.-P., I.J.C.P.-O. and L.R.V.P. All authors have read and agreed to the published version of the manuscript.

**Funding:** This research received no external funding.

**Institutional Review Board Statement:** Not applicable.

**Informed Consent Statement:** Not applicable.

**Data Availability Statement:** Not applicable.

**Conflicts of Interest:** The authors declare no conflict of interest.

## Appendix A. AddC Elements of the Fisher Matrix

The elements of the Fisher matrix with which the Hessian is calculated are presented in this appendix. Those elements are combined with the equations of Section 9 to obtain the estimates of AddC. The elements of the Fisher matrix based on Equation (22) are defined as:

$$J(\delta) = - \begin{bmatrix} I_{\alpha\alpha} & I_{\alpha\beta} & I_{\alpha\theta} & I_{\alpha\lambda} \\ I_{\beta\alpha} & I_{\beta\beta} & I_{\beta\theta} & I_{\beta\lambda} \\ I_{\theta\alpha} & I_{\theta\beta} & I_{\theta\theta} & I_{\theta\lambda} \\ I_{\lambda\alpha} & I_{\lambda\beta} & I_{\lambda\theta} & I_{\lambda\lambda} \end{bmatrix}. \tag{A1}$$

Each element of the matrix obtained in Equation (A1) can be obtained from the second partial derivation of Equations (23)–(26), which results in the following:

$$a = \ln(x_\iota)e^{x_\iota^\beta + 2x_\iota^\theta} x_\iota^{2\theta + 2\beta - 1} \beta\lambda\theta + x_\iota^{2\theta + \beta - 1}\theta\lambda(\beta\ln(x_\iota) + 1)e^{x_\iota^\beta + 2x_\iota^\theta}.$$

$$b = \alpha\left[(\beta\ln(x_\iota) + 1)x_\iota^{\theta - 1 + 2\beta} + x_\iota^{3\beta + \theta - 1}\beta\ln(x_\iota)\right]\beta\,e^{2x_\iota^\beta + x_\iota^\theta}.$$

$$c = \ln(x_\iota)\left[(3\beta\ln(x_\iota) + 2)x_\iota^{2\beta - 1} + (\beta\ln(x_\iota) + 2)x_\iota^{\beta - 1} + x_\iota^{3\beta - 1}\ln(x_\iota)\beta\right]e^{x_\iota^\beta}\alpha.$$

$$d = e^{x_i^\theta}\ln(x_i)\left((3\theta\ln(x_i) + 2)x_i^{2\theta - 1} + (\theta\ln(x_i) + 2)x_i^{\theta - 1} + x_i^{3\theta - 1}\ln(x_i)\theta\right)\lambda.$$

$$I_{\alpha\alpha} = -\beta^2 \sum_{\iota=1}^m \left[\frac{x_i^{2\beta}e^{2x_i^\beta}}{\left(\alpha\beta x_\iota^\beta e^{x_i^\beta} + \lambda\theta x_i^\theta e^{x_i^\theta}\right)^2}\right].$$

$$I_{\alpha\beta} = \sum_{\iota=1}^m \left[\frac{\lambda\theta(a + b)x_\iota}{\left(\alpha\beta x_\iota^\beta e^{x_\iota^\beta} + \lambda\theta x_\iota^\theta e^{x_\iota^\theta}\right)^3} - x_\iota^\beta\ln(x_\iota)e^{x_\iota^\beta}\right].$$

$$I_{\alpha\theta} = -\lambda\beta\left[\sum_{\iota=1}^m \left\{\frac{e^{x_\iota^\beta + x_\iota^\theta}\left(\ln(x_\iota)x_\iota^{\beta + 2\theta}\theta + \ln(x_\iota)x_\iota^{\beta + \theta}\theta + x_\iota^{\beta + \theta}\right)}{\left(\alpha\beta x_\iota^\beta e^{x_\iota^\beta} + \lambda\theta x_\iota^\theta e^{x_\iota^\theta}\right)^2}\right\}\right].$$

$$I_{\alpha\lambda} = -\beta\theta \left[ \sum_{i=1}^{m} \left\{ \frac{e^{x_i^\beta + x_i^\theta} x_i^{\beta+\theta}}{\left( \alpha\beta x_i^\beta e^{x_i^\beta} + \lambda\theta x_i^\theta e^{x_i^\theta} \right)^2} \right\} \right].$$

$$I_{\beta\alpha} = I_{\alpha\beta}.$$

$$I_{\beta\beta} = \sum_{i=1}^{m} \left[ \frac{c}{\alpha\beta x_i^{\beta-1} e^{x_i^\beta} + \lambda\theta x_i^{\theta-1} e^{x_i^\theta}} - \frac{e^{2x_i^\beta} \alpha^2 \left( x_i^\beta \ln(x_i)\beta + \ln(x_i) x_i^{2\beta}\beta + x_i^\beta \right)^2}{\left( \lambda\theta x_i^\theta e^{x_i^\theta} + \alpha\beta x_i^\beta e^{x_i^\beta} \right)^2} - e^{x_i^\beta} \ln(x_i)^2 \alpha \left( x_i^\beta - x_i^{2\beta} \right) \right].$$

$$I_{\beta\theta} = \sum_{i=1}^{m} \left[ \frac{e^{x_i^\theta + x_i^\beta} \left( x_i^{2\beta+2\theta} \ln(x_i)^2 \beta\theta + \beta\ln(x_i)(\theta\ln(x_i)+1) x_i^{\theta+2\beta} + (\beta\ln(x_i)+1)\left( \ln(x_i) x_i^{2\theta+\beta}\theta + x_i^{\theta+\beta}(\theta\ln(x_i)+1) \right) \right)}{\left( \lambda\theta x_i^\theta e^{x_i^\theta} + \alpha\beta x_i^\beta e^{x_i^\beta} \right)^2} \right].$$

$$I_{\beta\lambda} = -\alpha\theta \cdot \sum_{i=1}^{m} \left[ \frac{e^{x_i^\theta + x_i^\beta} \left( \ln(x_i) x_i^{\theta+\beta}\beta + \beta\ln(x_i) x_i^{\theta+2\beta} + x_i^{\theta+\beta} \right)}{\left( \lambda\theta x_i^\theta e^{x_i^\theta} + \alpha\beta x_i^\beta e^{x_i^\beta} \right)^2} \right].$$

$$I_{\theta\alpha} = I_{\alpha\theta}.$$

$$I_{\theta\beta} = I_{\beta\theta}.$$

$$I_{\theta\theta} = \sum_{i=1}^{m} \left[ \frac{d}{\alpha\beta x_i^{\beta-1} e^{x_i^\beta} + \lambda\theta x_i^{\theta-1} e^{x_i^\theta}} - \frac{e^{2x_i^\theta} \lambda^2 \left( \ln(x_i) x_i^\theta\theta + \ln(x_i) x_i^{2\theta}\theta + x_i^\theta \right)^2}{\left( \lambda\theta x_i^\theta e^{x_i^\theta} + \alpha\beta x_i^\beta e^{x_i^\beta} \right)^2} - \ln(x_i)^2 e^{x_i^\theta} \lambda \left( x_i^{2\theta} - x_i^\theta \right) \right].$$

$$I_{\theta\lambda} = \sum_{i=1}^{m} \left[ \frac{e^{x_i^\theta + x_i^\beta} \alpha\beta \left( \ln(x_i) x_i^{2\theta+\beta}\theta + \ln(x_i) x_i^{\theta+\beta}\theta + x_i^{\theta+\beta} \right)}{\left( \lambda\theta x_i^\theta e^{x_i^\theta} + \alpha\beta x_i^\beta e^{x_i^\beta} \right)^2} - x_i^\theta \ln(x_i) e^{x_i^\theta} \right].$$

$$I_{\lambda\alpha} = I_{\alpha\lambda}.$$

$$I_{\lambda\beta} = I_{\beta\lambda}.$$

$$I_{\lambda\theta} = I_{\theta\lambda}.$$

$$I_{\lambda\lambda} = \sum_{i=1}^{m} \left[ \frac{x_i^{2\theta} e^{2x_i^\theta}}{\left( \lambda\theta x_i^\theta e^{x_i^\theta} + \alpha\beta x_i^\beta e^{x_i^\beta} \right)^2} \right].$$

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
