# Peer review of "An Additive Chen Distribution with Applications to Lifetime Data"

_axioms, doi:10.3390/axioms12020118_

Round 1

Reviewer 1 Report

Attached.

Author Response

Dear Reviewer.

We ask you to review our response in the attached document. Changes made have been highlighted in blue within the manuscript

Reviewer 2 Report

This paper presents a model to estimate life, with properties to represent increasing, decreasing, and bathtub curve shapes for failure rates.

The Chen distribution was used as the basic model for the suggested model, which was constructed using the additive process, introducing the Additive Chen distribution (AddC).

Authors emphasize the exceptional flexibility of the AddC model in characterizing failure rates with non-monotonic behavior or with the shape of a bathtub curve in comparison to other existing models.

Here are the suggestions:

1.       It is not quite clear how other models represent survival time data with non-monotonic behavior. In lines 25-29 it is mentioned through the introduction that some authors are presenting methodologies that represent these cases. I strongly suggest a paragraph extending an explanation of the importance of this research in comparison with all these methodologies. Motivation could be improved adding just a simple paragraph. (Why is important the cases where non-monotonic behavior is present?)

2.       In lines 45-49, main objective is mentioned, nevertheless, I recommend to extend the importance of the model to describe non-monotonic behavior in real applications.

3.       Figure 4 needs to be edited, there is a cut in the graph.

4.       Figure 6 needs to be edited.

5.       Figure 8 needs to be edited.

In general, the authors should highlight the motivation of this paper. It is a good paper.

Author Response

(The authors gave the same response as above.)
